# The Effects of a High-Fat Diet on Inflammatory Bowel Disease

**DOI:** 10.3390/biom13060905

**Published:** 2023-05-30

**Authors:** Yuan Dang, Chunxiang Ma, Kexin Chen, Yiding Chen, Mingshan Jiang, Kehan Hu, Lili Li, Zhen Zeng, Hu Zhang

**Affiliations:** 1Department of Gastroenterology, West China Hospital, Sichuan University, Chengdu 610041, China; 2Centre for Inflammatory Bowel Disease, West China Hospital, Sichuan University, Chengdu 610041, China; 3Laboratory of Inflammatory Bowel Disease, Institute of Immunology and Inflammation, Frontiers Science Center for Disease-Related Molecular Network, West China Hospital, Sichuan University, Chengdu 610041, China

**Keywords:** high-fat diet, inflammatory bowel disease, intestinal barrier, intestinal microflora, pattern recognition receptor

## Abstract

The interactions among diet, intestinal immunity, and microbiota are complex and play contradictory roles in inflammatory bowel disease (IBD). An increasing number of studies has shed light on this field. The intestinal immune balance is disrupted by a high-fat diet (HFD) in several ways, such as impairing the intestinal barrier, influencing immune cells, and altering the gut microbiota. In contrast, a rational diet is thought to maintain intestinal immunity by regulating gut microbiota. In this review, we emphasize the crucial contributions made by an HFD to the gut immune system and microbiota.

## 1. Introduction

Inflammatory bowel disease (IBD), including ulcerative colitis (UC) and Crohn’s disease (CD), causes significant gastrointestinal symptoms, such as diarrhea, abdominal pain, bleeding, and weight loss. IBD is associated with a spectrum of extraintestinal manifestations, including sclerosing cholangitis, uveitis, arthritis, erythema nodosum, and pyoderma gangrenosum [1]. The steadily increasing emergence of IBD has cause great burden to the health care system and society worldwide; therefore, it is important to explore its underlying pathogenesis. Although identifying its exact pathogenesis remains difficult, overwhelming data suggests that its prevalence is related to Western diets [2] which feature high saturated fat, high sugar (HS), and low fiber exposure [3,4]. Similar to Western diets, a high fat diet (HFD) contains high amounts of fatty acids and is low in fiber and vitamins [5]. This dietary shift has been proposed as an etiological factor of IBD because the accelerating incidence of IBD in newly industrialized countries is in line with the westernized dietary structure [6,7,8]. The severity of IBD is also influenced by hormonal regulation under food intake [9]. For example, ghrelin is not only a multi-faceted gut hormone that regulates gastrointestinal motility, stimulates food intake, and promotes fat deposition [10]; it is also a mediator to prevent intestinal inflammation [10,11]. Research demonstrates that ghrelin supplements can alleviate inflammation in different models of IBD or accelerate the healing of gastrointestinal ulceration [12,13,14,15,16,17], while consequently a high fat diet could decrease levels of ghrelin and impair its anti-inflammatory effect [18]. Additionally, HFD can reduce the secretion of somatostatin and increase the level of serotonin [19,20], resulting in the exacerbated inflammation of IBD [20,21]. Therefore, to adopt appropriate therapies to treat IBD, it is imperative to uncover the mechanisms between diet and IBD and their potential implication in reducing IBD morbidity, achieving IBD remission, and reducing the disease burden on patients and healthcare systems.

Intestinal immunity, as a pivotal factor in IBD pathogenesis, cannot be underestimated in IBD progression because of its unique roles in warding off pathogen invasion from the gut and retaining host–microbial immune homeostasis [22]. The dysfunction of intestinal immunity may lead, for various reasons, to increased intestinal permeability and makes it easier for pathogens and their products to enter host tissues, thus resulting in worsening conditions [23]. Therefore, increased intestinal permeability has significance in predicting disease recurrence in IBD [24,25]. Moreover, the dysregulation of adaptive immunity is also a predictor of the clinical course of IBD [26], and genetics are also essential participants in these immune processes [27]. These studies conclude that intestinal immunity is intimately involved in the pathogenesis of IBD. The intestinal microbiota has been linked to the development of IBD for many years. *Firmicutes* and *Bacteroidetes* are the predominant bacteria in the human intestine; however, a lower proportion of these species was found in IBD patients, both in children and adults [28]. Notably, these two obligate anaerobes are responsible for catabolizing fiber into short-chain fatty acids. A reduction in their abundance would reduce the production of short chain fatty acids and result in IBD occurrence or accelerate the IBD course due to the loss of their anti-inflammatory properties [29,30].

Additionally, an increase in adherent–invasive E. Coli (AIEC), which plays a pathogenic effect on disease development with the ability to adhere and invade intestinal epithelial cells [31], was also displayed in CD patients. The role of gut microbiota composition and function in the development and severity of IBD is demonstrated by the changes in microbiota composition among IBD patients observed in these studies. Engineering the microbiota as an adjunct to boost current IBD therapies would be a promising approach.

Because of the meaningful roles of diet, intestinal immunity, and microbiota in IBD, many studies have attempted to explore their links to the three entities in IBD (Figure 1). 

In this review, we will focus on the critical roles of a high-fat diet (HFD) on intestinal immunity and gut microbiota and the impacts of the altered intestinal microecological imbalance on intestinal immunity in IBD, aiming to offer a direction for the pathogenesis of IBD (Figure 2).

## 2. High-Fat Diet and the Intestinal Barrier in IBD

Intestinal epithelial cells (IECs) and tight junctions (TJs) constitute the epithelial barrier of intestinal immunity; its integrity is the precondition for intestinal defense [32]. Mucosal defense, as the front of the epithelial barrier, including immune molecules and mucus proteins, represents the most prominent interaction zone between the microbiota and host [33]. The receptors expressed in the intestinal epithelial cells also play vital roles in the constant communication of gut microflora and immune cells aggregated in the mucosal epithelium or submucosal lamina propria. Various studies have shown that an abnormal intestinal barrier function is crucial to the pathogenesis of IBD [34,35,36]. Moreover, HFD was shown to participate in intestinal inflammatory responses by affecting the intestinal barrier function [37] (Figure 2).

### 2.1. TJs

TJs are multifaceted structures composed of multiple components, such as claudin, occludin, and zona occludin-1 (ZO-1), whose integrity can be measured by transepithelial electrical resistance (TEER) [38]. In a rat model of acute pancreatitis, TJ distribution may be altered by HFD, enhancing the entrance of gut microorganisms and rendering the host susceptible to IBD [39]. The mRNA level of intestinal TJ proteins, such as ZO1 and occludin, could be directly decreased by saturated fat in HFD, thereby increasing intestinal permeability in mice with HFD-induced obesity [40]. Consistently, Oliveira et al. [41] treated Caco-2 and MDCK epithelial cell lines with the intestinal luminal content separated from high-fat-fed mice and found that these epithelial barrier structures could be altered in vitro. In another in vitro test, the standard component of the HFD to deal with Caco-2 cell monolayers was used directly. It was found that HF concentration had a positive correlation with the aluminum absorption through the paracellular route, whereas it maintained the opposite association with TEER [42]. An associated mechanism has been noted in which fatty acids could undergo a complexing reaction with calcium ions (Ca2^+^), which is essential for the normal functioning of TJ protein molecules [43].

Hydrophobic bile acid (BA), the product of HFD, can disrupt intestinal TEER through the reactive oxygen species (ROS)-mediated phosphoinositide 3-kinase (PI3K) pathway [44]. Consequently, it results in the tyrosine phosphorylation of ZO1 and β catenin and, subsequently, prevents the binding of E-cadherin with other TJ structures in vitro [44]. Furthermore, in mice with HFD-induced obesity, deoxycholic acid (DCA), as the primary component of BA, was found to decrease occludin expression and switch the protective claudin protein into the leaky claudin protein in the intestine, claudin-3, and claudin-2. The potential mechanism of claudin-2 upregulation by DCA has been related to the tumor necrosis factor-α (TNF-α)/nuclear factor kappa-B (NF-κB)/mitogen-activated protein kinase (MAPK) signaling pathway [45]. In an in vitro test, DCA has also been shown to promote occludin dephosphorylation and cytoskeletal rearrangement at the TJ level through the epidermal growth factor receptor (EGFR)-mediated autophosphorylated signaling pathway, exerting a direct detrimental impact on intestinal permeability [46]. This mechanism of action of the HFD has been further confirmed by the result that this estimated detrimental effect could be reversed by EGF application [46]. However, high-fiber diet-induced changes are beneficial for TJ proteins. In a clinical trial, feeding a diet supplemented with soluble fiber increased plasma glucagon-like peptide 2 (GLP-2) levels in a time-dependent manner [47]. GLP-2 has been reported to influence the appearance of ZO-1, occludin, and claudin-1 proteins in intestinal tissue via the MAPK-ERK1/2 signaling pathway. This signaling pathway involves TJ protein expression and intestinal permeability [48]. The higher expression of occludin and claudin in the colon was also found in fiber-treated mice with dextran sulfate sodium (DSS)-induced colitis [49]. In conclusion, these results support the idea that a combination of low fatty acids and higher soluble fiber benefits the intestinal barrier by modifying the TJ structure.

### 2.2. Mucin 2

Mucin 2 (Muc2), a critical component of the mucosal layer, is a vital component of innate intestinal immunity and is the main product of goblet cells, which differentiate from multipotent stem cells positioned at the bottom of the crypts [50]. Results found by Hussain et al. [51] show that the crypt structure and goblet cells in the colon of HFD-fed mice may be directly destroyed by HFD to reduce Muc2 abundance at the histopathological level, downregulate Muc2 mRNA at the transcriptional level, and reduce interleukin-22 (IL-22) in the colon and serum. Conspicuously, IL-22 should seemingly be recognized as an underlying molecular mechanism of Muc2 reduction. A mouse model of Th2-mediated colitis supported the theory that Muc production could be enhanced by IL-22 [52]. Strikingly, IL-22 administration could reverse the intestinal inflammation stimulated by an HFD and improve mouse body weight in early life [53]. The signaling pathway can be further explained by a higher expression of IL-22 activating STAT3 in epithelial cells, which protects against intestinal inflammation and promotes mucosal healing [54,55].

Furthermore, Muc2 secretion in DSS-induced mouse colitis is positively regulated by the peroxisome proliferator-activated receptor-gamma (PPAR-γ) via the PPAR-γ/long myosin light chain kinase (MLCK)-dependent pathway [56]. The dysregulated PPAR-γ pathway in the case of the HFD-fed group of mice could be accounted for by the decreased expression of Muc2 in contrast to the standard diet group [57]. However, what needs to be emphasized is that IL-22 is a double-edged sword in IBD; its complex roles in intestinal inflammation depend on the cytokine environment, disease type, and disease location. Concretely speaking, the constitutive expression of IL-22 in the small intestine is a protective factor for epithelial barrier integrity [58,59]. Still, its deleterious roles are found in large bowels, and CD is more influenced by the expression level of IL-22 than UC [52,60]. The potential mechanism could be ascribed to the fact that the source of IL-22 varies among many cell types and the cells in the corresponding tissue [61]. Thus, the above influencers should be considered when exploring the exact regulatory mechanisms of the HFD on Muc2 production in IBD. Indeed, the mucin family contains a series of members, and the distinctive roles of HFD on other mucin secretions and the specific regulating signal pathways are waiting to be described.

### 2.3. AMPs

Antimicrobial peptides (AMPs) are critical components of the intestinal mucosal layer, and their defects contribute to intestinal inflammation and drive IBD occurrence [62]. The influence of an HFD on AMP secretion has been demonstrated in numerous studies. A reduction in AMP production was observed in HFD-fed mice, including a decrease in lysozymes, Reg-IIIγ, and angiogenin 4 [63]. In terms of gene expression levels, not only was the expression of a series of AMP-coding genes significantly downregulated by the HFD, but the level of the gene responsible for AMP maturation, matrix metalloproteinase-7, was also negatively modulated by the HFD. Interestingly, these gene expression levels are consistent with PPAR-γ gene expression, thus indicating that the PPAR-γ pathway is responsible for the effects of the HFD on AMP secretion [57]. At the transcriptional level, a decrease in Trefoil factor 3 (TFF3) mRNA, the activator of intestinal defensins, was observed in mice that were fed a Western diet [64]. Moreover, the secreted AMPs from Paneth cells are primarily dependent on IL-22. Thus, AMPs are simultaneously affected by the IL-22 and PPAR-γ pathways, such as MUC2 secretion, in Western diet patterns.

## 3. High-Fat Diet and Pattern Recognition Receptors in IBD

Pattern recognition receptors (PRRs) are immune sensors that recognize pathogen-associated molecular patterns (PAMPs) inside and outside the cells. Many PRRs, such as Toll-like receptors (TLRs) and NOD-like receptors (NLRs) and the molecules interacting with them, have been identified and characterized. PRRs are essential in the detection and elimination of pathogens. However, dysregulated PRR activation can lead to inflammatory diseases. It has been shown by increasing evidence that innate immune dysfunction, which is mediated by TLRs and NLRP3, plays a crucial role in the pathogenesis of IBD [65,66]. Previous research has shown that PRR-mediated inflammation, which is caused by an HFD, plays a notable role in intestinal inflammation [67,68,69,70,71,72] (Figure 3).

### 3.1. TLRs

TLRs are the main intracellular PRRs that recognize the corresponding pathogen-associated molecular patterns (PAMPs) in bacteria. Once the two molecules interact with each other, bacterial invasion signaling is received by myeloid differentiation factor 88 (MyD88). Subsequently, NF-κB p65, the anchor protein for NF-κB, dissociates from its inhibitor IκB. Then, the transcription factor NF-κB is translocated from the cytoplasm to the nucleus, and the production of downstream proinflammatory cytokines is switched on, resulting in intestinal inflammation [73]. An HFD can modulate this process and disturb gut immunity to induce inflammation. Several rat models of obesity have demonstrated that the TLR4/NF-κB/MyD88 pathway is involved in the inflammatory process by upregulating the levels of cytokines, such as TNF-α, IL-1β [74], IL- 4, IL-6, IL-17, and IL-18, in the colon [75]. Moreover, the inhibition of TLR4 by TAK242 in a high-fat diet-fed mouse model can reduce intestinal inflammation [67]. Intriguingly, TLR4 has dual effects in colitis, maintaining mucosal integrity while also intensifying colitis by promoting proinflammatory cytokine release [65]. Considering the increased expression of TLR4 in CD and UC patients, TLR4 might give rise to ascending intestinal inflammation and disease development [75].

In addition to proinflammatory factors, TLR4 can regulate cellular oxidative stress expression and lead to intestinal inflammation [76]. The oxidative stress markers nicotinamide adenine dinucleotide phosphate (NADPH), NADPH oxidase 4 (NOX4), and NOX1 were considerably increased in the mouse ileum upon HFD consumption [68]. Moreover, inducible nitric oxide synthase (iNOS) and cyclooxygenase-2 (COX-2) in the mouse colon were also increased in the HFD group when compared with the low-fat diet group by phosphorylating Akt-regulated forkhead transcription factor (FOXO) [69]. Intriguingly, in the intestinal mucosa of mice that were administered oxidized fat, which is a component of fried food, the oxidative stress-responsive transcription factor nuclear factor-erythroid 2-related factor 2 (Nrf2) was upregulated [77]. At the same time, the protective role of Nrf2 in colitis to control antioxidative enzymes and proinflammatory cytokines was proven by multiple studies [78,79,80]. In addition to the TLR4 receptors, HFD increased fecal deoxycholic acid (DCA) production in the mouse colon to activate TLR2, promoting M1 macrophage polarization and proinflammatory cytokine production through the M2 muscarinic acetylcholine receptor (M2-mAchR)/Src pathway [70]. In a clinical trial, the dysfunction of TLR2 was observed in young men who had been given an HFD, which may contribute to the increased risk of infection in diet-induced obesity [81]. In conclusion, TLRs are indispensable for HFD-induced intestinal inflammation, and other TLR channels need additional exploration to reveal their roles in colitis.

### 3.2. NLRs

NLRs are another type of intracellular PRR involved in innate intestinal immunity. NLRP3 is the most studied NLR and has been proven to be mediated in DSS-induced colitis by regulating IL-1β secretion [82,83]. Moreover, polymorphisms in NLRP3-related genes are associated with susceptibility to IBD [84]. Responding to the signaling of pathogenic invasion, NLRP3 will connect with its downstream adaptor protein, i.e., the apoptosis-associated speck-like protein (ASC) containing a caspase recruitment domain (CARD). Concomitantly, CARD interacts with its downstream caspase-1 and cleaves its substrates pro-IL-1β and pro-IL-18 into biologically functional forms, i.e., the proinflammatory cytokines IL-1β and IL-18, respectively [84,85]. The NLRP3 inflammasome is widely expressed in immune cells, including dendritic cells (DCs), macrophages, and monocytes [86]. HFD can activate the NLRP3 inflammasome in these immune cells. An increased infiltration of DCs primed by the HFD challenge was found in the adipose tissue of C57BL/6 mice [71]. In this study, higher levels of proinflammatory cytokines were secreted by isolated HFD-derived bone marrow-derived DCs (BMDCs) as compared with chow-derived DCs; more maturation markers of DCs, CD40, CD80, and CD86 were expressed as well as NLRP3 inflammasome members, such as IL-1RI, caspase-1, TLR4, and NLRP3 mRNA. The increased secretion of IL-1β occurred dose-dependently when BMDCs were treated with palmitic acid; this effect disappeared in TLR4-deficient BMDCs [71]. In a human monocytic cell line, an HFD was again revealed to induce NLRP3 expression and promote the secretion of pro-IL-1β and IL-1β through TLR2 activation. Similarly, this role was inhibited by knocking down TLR2 [72] (Figure 2). Therefore, HFD-induced NLRP3 inflammasome expression was dependent on TLRs (Figure 3). 

In addition to the mentioned in vitro experiments, in vivo experiments further indicated that HFD exposure leads to intestinal inflammation through NLRP3 inflammasome activation. In an animal study, increased NLRP3 inflammasome, TNF-α, IL-1β, and macrophage infiltration were detected in colon tissues in the HFD-treated group [72]. Moreover, leukocyte accumulation in the zebrafish intestine can occur secondary to high cholesterol activating inflammasomes, creating local and systemic inflammation; NF-κB and TNF-α are involved in this process [87]. Although the NLRP3 inflammasome has been reported to induce intestinal inflammation by secreting the proinflammatory cytokines IL-1β and IL-18 in early studies, conflicting results may be rereviewed for the protective roles of NLRP3 in DSS-induced intestinal damage [84]. It has been shown that mesenchymal stem cells (MSC) primed by IL-1β could alleviate intestinal inflammation in DSS-induced colitis. [88]. Indirectly, IL-1β induces the production of prostaglandin E2 (PGE2), which has been shown to inhibit mucosal inflammation in mice with DSS-induced colitis [89,90]. Therefore, the complicating effects of the NLRP3 inflammasome pathway on IBD under an HFD need further study.

## 4. High-Fat Diet and Immune Cells in IBD

The disordered function and the number of adaptive immune cells, especially T lymphocytes, are crucial to IBD etiology [91,92,93,94] (Figure 2). A regulatory role of HFD in adaptive immune cells was also demonstrated in HFD-fed DSS-induced colitis mice; increased non-CD1d-restricted NKT cells and decreased Tregs were observed in the colon. This is paralleled by the fact that a higher ratio of effector T cells/regulatory T cells will contribute to the inflammatory state in the intestine [95]. Moreover, HFD can disrupt the steady state of intraepithelial lymphocytes (IELs), reducing homeostatic proliferation in intraepithelial T lymphocytes and the expression of CD103 and CCR9 on these cells to worsen the outcome of DSS-induced colitis in mice [96]. Similarly, HFD-fed mice substantially increased Th17 polarization [97] and CD3+ T cell infiltration in the gut [98], resulting in colitis aggravation. HFD-produced peroxidized lipids, such as 13-HPODE, could stimulate the secretion of pro-inflammatory granzymes by resident NK cells, thereby contributing to intestinal inflammation [99]. Previous studies of IBD suggest the involvement of innate lymphoid cells (ILCs) [100]. There was an increase in IL-17-producing type 3 innate lymphoid cells (ILC3s) in the offspring of mice that were fed a high-fat diet, leading to high susceptibility to inflammation [101]. In another animal study, significantly increased expression of caspase-3 was found in type 1 innate lymphoid cells (ILC1) and ILC3 in the mice fed with HFD as compared to the control group, possibly resulting in pro-apoptotic mechanisms [102]. In terms of B cells, in the plasma and spleen of C57BL/6 mice, the HFD induced evident decreases in the number of B cells, accompanied by oxidative stress and increased oxidative damage [103]. Gurzell et al. [104] found that docosahexaenoic acid (DHA), a type of n-3 polyunsaturated fatty acid that is low in Western diets, enhances B cell activation, thereby boosting the humoral immunity of mice, which may up-regulate the resolution phase of inflammation.

Furthermore, an increased presence of colonic macrophages has been shown in HFD-induced obese mice [105]. Still, they are deficient in forkhead box O3 (FOXO3) in macrophages [105], which mediates the proapoptotic or anti-inflammatory effects of macrophages [106]. Neutrophils, generally regarded as keys to inflammation, play a crucial role in intestinal inflammation in IBD [107]. Neutrophil migration is enhanced by an HFD due to the elevated expression of associated cytokines [108], such as monocyte chemoattractant protein-1 (MCP-1) [109] and chemokine (C-X-C motif) ligands 1 (CXCL1) and 2 (CXCL2), in the intestine of mice [110]. Yoshida et al. [111] found that the secretion of growth-regulated oncogene/cytokine-induced neutrophil chemoattractant-1 (GRO/CINC-1) could be increased by long-chain fatty acids in rat IECs.

In a mice model of Crohn’s disease-like ileitis, DCs recruited into the intestinal lamina propria also appear because of the enhanced levels of chemokine (C-C motif) ligand (CCL) 20 and intercellular cell adhesion molecule 1 (ICAM1) induced by the HFD [112]. Moreover, increased maturity markers of DCs were found in a DSS-induced colitis model that was fed an HFD, which exacerbated intestinal inflammation [71,113].

All these data illustrate that an HFD could influence intestinal immune cells and the humoral immune response to induce intestinal inflammation, disrupt tissue structure, and exacerbate IBD conditions. More emphasis should be placed on the fact that some studies have concentrated on the changes in these cytokines and immune cells under an HFD but have not explored the alterations in intestinal microorganisms. Whether these immunity-related changes are just the results of this dietary model, the results of intestinal microorganisms, or both, is still obscure. Therefore, it is necessary to conduct these studies in germ-free mice, which may better elaborate on the direct effect of diet on intestinal immunity and help explain how environmental factors can increase susceptibility to IBD.

## 5. Polyunsaturated Fatty Acids (PUFAs) in IBD

Polyunsaturated fatty acids (PUFAs) are a type of fatty acid that contains more than two double bonds. PUFAs contain two principal families: n-6 (or omega-6) and n-3 (or omega-3). Eicosapentaenoic acid (EPA) and DHA are precursors of n-3 PUFAs and are classified as essential lipid mediators. The typical Western-style diet has a high n-6/n-3 ratio of approximately 10–15:1 [114]. The increase in dietary n-6/n-3 PUFAs was positively correlated with the increased incidence of IBD [115] (Figure 2). John et al. [116] suggested that increasing the consumption of n-3 PUFAs may help prevent UC. In addition, prospective research has revealed that increasing the proportion of n-3/n-6 PUFA ingestion can help to maintain IBD remission [117].

Beguin et al. [118] showed that 150 mM of DHA could increase ZO-1 intensity in vitro. A lower intensity of occludin, when incubated with n-6 PUFAs, was also found in this research. Previous research has shown that TLR-2 gene expression in TNBS-induced colitis may be promoted by n-3 PUFAs [119]. The effect of n-3 PUFAs on neutrophils in the inflammatory process has also been investigated in TNBS mice in vivo; DHA and EPA could inhibit PMN transepithelial migration by reducing VCAM-1 and ICAM-1 [119]. Studies have revealed that the serum level of leukotriene B4 (LTB4) secreted by neutrophils in UC patients could be reduced by n-3 PUFAs and that the extravascular tissue damage caused by excessively activated neutrophils could also be avoided [120]. Current studies in mice show that n-3 polyunsaturated fatty acids could reduce the antigen-presenting function of DCs by inhibiting the expression of CD69 and CTLA-4 on T lymphocytes [121,122]. The metabolites of linoleic acid (LA, n-6 PUFAs) and arachidonic acid (AA, n-6) can produce thromboxane B2 (TXB2) and 4-series leukotrienes (LTS) through the cyclooxygenase (COX) pathway [123]. Additionally, n-6 PUFAs could also promote the production of PGE2 in vitro, thereby increasing the production of costimulatory molecules, including OX40 and CD70 in both DCs and T cells; this could induce T-cell proliferation and cause proinflammatory effects [124,125]. Moreover, the role of n-3 PUFAs in distinct types of enteritis models seems to differ. In the chronic model of intestinal inflammation, higher levels of suppressive cytokines are expressed by Th17 cells in the colon of mice than in the spleen. At the same time, there was no difference in the acute model [126,127]. Therefore, the relationship between PUFAs and T cells, especially Th17 cells in IBD, requires further study.

## 6. Short Chain Fatty Acids (SCFAs) in IBD

SCFAs are the products of anaerobic fermentation of dietary fiber, primarily containing acetate, propionate, and butyrate [8,128]. Nowadays, the beneficial roles of SCFAs on intestinal barrier integrity and immune cell functions in IBD have been highlighted by increasing evidence. Zheng et al. [129] revealed that claudin-2 formation is negatively regulated by butyrate via upregulating IL-10RA expression to protect gut barrier function, which was related to the signal transducing activator of transcription 3 (STAT3)-Histone Deacetylase inhibition (HDACi) pathway. Moreover, Hatayama et al. [130], who treated the human colon cancer cell line with butyrate and confirmed that SCFAs stimulates MUC2 production both in protein and mRNA levels, revealed that SCFAs increase MUC2 production. 

The mechanism of butyrate stimulating AMP production has been investigated by Zhao et al. [131]. They used mammalian target of rapamycin (mTOR) siRNA and STAT3 siRNA to knockdown mTOR and STAT3, respectively, in intestinal epithelial cell (IEC) models and found that the mRNA and protein expressions of RegIIIγ and β-defensins were prominently impaired in these IECs, thus indicating that butyrate could active mTOR and STAT3 to promote AMP synthesis and confer resistance to colitis. The STAT3 and mTOR pathways have also been demonstrated to promote Th1 cells producing IL-10 by using butyrate to deal with the T cells from IBD patients and the DSS model, therefore limiting colitis [132]. Noteworthy, the role of SCFAs on T cells is related to the cytokine milieu. In the case of SCFA treatment, effector T cells, including Th1 and Th17 cells, are generated from naive T cells under a steady inflammatory condition, whereas in the active immune responses, the productions are regulatory T cells, such as IL-10+ T cells and FoxP3+ T cells [133]. Meanwhile, DC differentiation has also been reflected to be associated with mTOR and SATA3 pathways [134]. Butyrate, as a HDAC3 inhibitor, increases the antimicrobial functions of intestinal macrophages through a reduction in mTOR kinase activity [135] and downregulates macrophages secreting proinflammatory mediators [136]. The effect of SCFAs on neutrophils in the inflammatory process has also been investigated both in rats vivo and in vitro. On the one hand, SCFAs can promote neutrophils recruited into inflammatory sites by increasing L-selectin expression and chemokine release [137]. On the other hand, pro-inflammatory cytokines, TNF-α, and NO, produced by lipopolysaccharide (LPS)-stimulated neutrophils, are inhibited by SCFA treatment [138]. In view of the fact that mTOR is crucial in regulating the differentiation and function of innate and adaptive immune cells for intestinal immunity and that STAT3 has prominent aspects in the expressions of cytokines and chemokines, great insights must be explored about the interrelations between immune homeostasis and SCFAs in IBD.

## 7. High-Fat Diet and Intestinal Dysbacteriosis in IBD

It is well acknowledged that IBD is linked to compositional and metabolic alterations in intestinal microbiota. Fecal microbiota transplantation has been reported as a potential treatment of IBD [139,140]. Microbial communities can speedily and flexibly convert their components and functional repertoires following modern dietary challenges. Examples of this conversion could be supported by studies showing that a high HFD intake would increase the abundance rates of AIEC and *Clostridioides difficile*, decrease the abundance rates of *Akkermansia muciniphila*, and reduce the abundance rates of both *Firmicutes* and *Bacteroidetes* [141]. The clinical trial devised by Fritsch et al. [142] revealed that UC patients treated with a low-fat, high-fiber diet experienced a reduction in inflammation and an increase in the abundance of Bacteroidetes. Several studies have shown that these gut microbiotas and their metabolites are strongly correlated with IBD by potentially affecting intestinal immunity (Table 1) (Figure 2).

### 7.1. AIEC

AIEC colonization is enhanced under Westernized diets, and their overgrowth could impair intestinal barrier integrity, alter the formation of the host mucus layer, and damage immune function in IBD. Carcinoembryonic antigen-related cell adhesion molecule 6 (CEACAM6) in the intestine is a vital receptor for AIEC invasion. Overexpressed CEACAM6 has been reported in the ileal mucosa of IBD patients, especially in the ileal mucosa [158]. Interestingly, the abnormal expression of claudin-2 in AIEC-infected CEABAC10 transgenic mice indicates that AIEC could target claudin-2 to exacerbate intestinal physical barrier dysfunction, ultimately inducing disease recurrence in CD patients [158]. This facilitation effect of AIEC on claudin-2 via the CEACAM6 receptor has also been shown in CEABAC10 mice fed an HF/HS diet [143].

Regarding the mucus barrier, the level of Muc-2 protein and the mRNA of its gene are decreased in the colonic tissue of CEABAC10 mice on an HF/HS diet compared with a conventional diet [143]. Such results could be ascribed to the fact that AIEC could directly or indirectly increase TNF-α release and activate the TNF-α-NF-κB regulatory pathway to impact cytoskeletal contraction and compromise intestinal permeability [144]. Furthermore, the AIEC-related flagellate receptors TLR5 and NOD2 were found to be upregulated in CEABAC10 mice that were fed a Western diet [143]. Increased expression of these PRRs in innate immunity has been shown in IBD to promote TNF-α synthesis and activate the inflammatory response, leading to intestinal inflammation. In addition to regulating pattern recognition receptors, AIEC can induce Th17 polarization and IL-17 release in the colon of mice to participate in mucosal immunity, requiring the virulence-associated metabolic enzyme propanediol dehydratase [145]. AIEC can activate the release of the proinflammatory cytokine IL-8 and chemokine CCL20 to impact the recruitment of dendritic cells and macrophages in the intestine of CD patients [146]. Combined with the abovementioned studies, a vicious cycle of intestinal barrier disruption has formed because a Western diet could directly increase AIEC invasion by triggering low-grade intestinal inflammation. In turn, such an inflammatory status can exacerbate AIEC invasion. Thus, a vital role in HFD-induced intestinal inflammation is played by the colonization of AIEC.

### 7.2. A. muciniphila

*A. muciniphila* is an anaerobic gram-negative bacterium capable of mucin degradation, and its existence contributes to regulating the intestinal mucosal layer. *A. muciniphila* in animal and cell models has been demonstrated to boost the gene expression of TLR2, TLR4, occludin, and claudin3. It has also been shown to promote the thickness of the mucosa in many studies, even in HFD feeding [151,152,153,154]. Supplementation with *A. muciniphila* has also been reported to enhance the production of the antimicrobial peptide Reg3γ in both the colon and ileum of HFD-fed mice [153]. Kosciow et al. [154] focused on the anti-inflammatory effect of *A. muciniphila* on the intestinal immune status by investigating a series of colonic gene expression profiles related to intestinal immunity and inflammation. In this study, immunoglobulin-encoding genes and receptors, chemokines CXCL13 and CCL12, and complement factors C1ra and C5ar1 in the mouse colon were inhibited by *A. muciniphila* when compared with the control group. Additionally, the presence of B cells was reduced in the colon after the mice were supplemented with *A. muciniphila*. In contrast, anti-inflammatory Treg cells proliferated with *A. muciniphila* administration in another study [155]. Antigen-specific Th cell responses in the gut of mice have also been induced by *A. muciniphila* [156]. Both anti- and proinflammatory cytokines, IL-1β, IL-6, IL-8, IL-10, and TNF-α, were upregulated under *A. muciniphila* in vitro [152]. Thus, although the overall trend from the current studies illustrated that *A. muciniphila* could protect the intestinal barrier function of the host, vigilance is still required because the complex role of *A. muciniphila* cannot be categorically classified as either anti- or proinflammatory.

### 7.3. Clostridioides difficile

*Clostridioides difficile* infection (CDI) has been well documented to be associated with relapse and therapeutic response in IBD. Two major toxins, *Clostridium difficile* toxins A and B (TcdA and TcdB) and *C. difficile* transferase (CDT), determine its pathogenicity; experiments of mice models and human intestinal organoids have demonstrated that all could cause inactive GTPases, ultimately leading to cytoskeleton disruption, cell death, and barrier breakdown [147,148]. Furthermore, toxins from CDI could result in the acceleration of intestinal epithelial stem cell (IESC) niche degeneration, defects in IESC regeneration, and disruption of the intestinal barrier [148]. With the production of significant hypervirulent strains, CDT could interact with the lipolysis-stimulated lipoprotein receptor (LSR) in the intestine, which is associated with the formation of tricellular tight junctions [159,160]. TcdB was capable of inducing epithelial cell death in an enzyme-independent manner by the ROS and NADPH oxidase complex in an experiment of colonic explants [161]. In addition to affecting the intestinal barrier, CDI has been shown to mediate immune responses in ASC-deficient and IL-1 receptor antagonist-treated mice. Ng et al. [149] first prescribed macrophages with TcdA and TcdB to stimulate interleukin (IL)-1 secretion and trigger intestinal injury by activating inflammasomes. In contrast, this facilitation of intestinal inflammation disappeared when these toxins were used to stimulate macrophages deficient in inflammasome components. In terms of cytokines and immune cells, it was demonstrated in a series of earlier studies that IL-8 synthesis could be induced by TcdA through the upregulation of NF-κB and mitochondrial oxygen radicals in colonocytes, thus causing neutrophil chemotaxis, monocyte necrosis, and colonic inflammation [162,163]. Moreover, CDT has been shown to decrease neutrophil recruitment by stimulating NOD-like receptors in the colon of mice [164]. Another animal study by Ryan et al. [150] has suggested that dendritic cells (DCs) and T helper (Th) cell responses may be modulated by the surface layer proteins (SLPs) isolated from CDT by recognizing TLR4. This research helped to better understand how CDI contributes to IBD exacerbation. To date, the mechanisms through which CDI injures the intestinal barrier have been elegantly characterized. A consequent challenge is identifying the definitive receptor for these toxins. Attempts to unveil the receptors will be valuable for therapeutic intervention of CDI-related IBD aggravation and recurrence.

## 8. Conclusions

Diet, intestinal immunity, and microbiota are interlocking and complex networks in IBD. On the one hand, an HFD disrupts the intestinal TJ structure; decreases intestinal TEER; weakens MUC2 and AMP secretion; actives the TLR/NF-κB and oxidative stress pathways; actives the NLRP3 inflammasome; and promotes intestinal immune cell infiltration, thereby bolstering intestinal inflammation. The HFD could also cause intestinal microecological imbalance. The damaged intestinal microecology acts on the above links to exacerbate intestinal inflammation and IBD-related intestinal damage. Thus, a vicious cycle between intestinal immunity and microbiota imbalance is caused by an inappropriately composed diet model. The administration of proper dietary prescriptions should be considered as a promising treatment for IBD patients. 

## Figures and Tables

**Figure 1 biomolecules-13-00905-f001:**
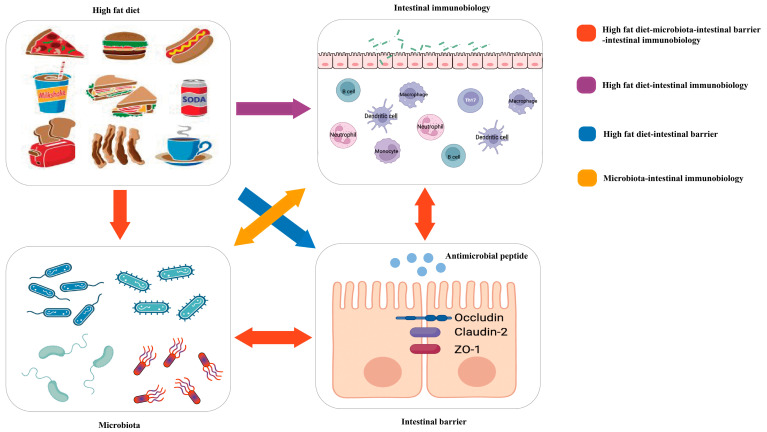
Red arrow: the high-fat diet shapes the gut microbiota, which changes the function of the intestinal barrier and further influences intestinal immunobiology. The intestinal immunobiology also affects the intestinal barrier and IECs, i.e., the components of the intestinal barrier, which also influence microbiota. Purple arrow: the high-fat diet modulates immune cell infiltration and polarization. Blue arrow: the high-fat diet influences the intestinal barrier. Orange arrow: gut microbiota interacts with intestinal immunobiology.

**Figure 2 biomolecules-13-00905-f002:**
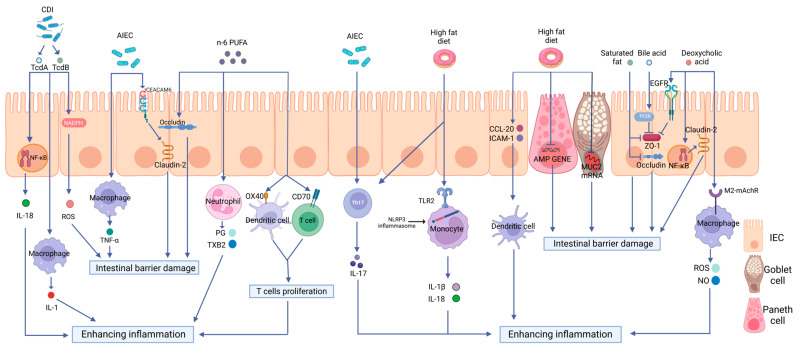
The effects of a high-fat diet on intestinal immunity in IBD. The specific manifestations are as follows: (1) Intestinal barrier dysfunction. A high-fat diet (HFD) disrupts the intestinal tight junction (TJ) structure and reduces Mucin 2 abundance and antimicrobial peptide (AMP) secretion. (2) Dysregulated PRR activation. HFD activates Toll-like receptor 2(TLR2) and consequently activates the NLRP3 inflammasome and promotes the release of inflammatory mediators, such as interleukin-18 (IL-18) and interleukin-1 beta (IL-1β), through monocytes to enhance intestinal inflammation. (3) The disordered function of immune cells. HFD can increase the production of OX40 and CD70 in both DCs and T cells, therefore inducing T cell proliferation and causing a proinflammatory effect. HFD can also stimulate the maturation of DCs, the polarization of Th17 cells, neutrophil migration, and the release of inflammatory mediators. (4) Intestinal dysbiosis. HFD can increase the abundance rates of AIEC and *Clostridioides difficile*, causing intestinal barrier disruption and enhancing intestinal inflammation via multiple mechanisms.

**Figure 3 biomolecules-13-00905-f003:**
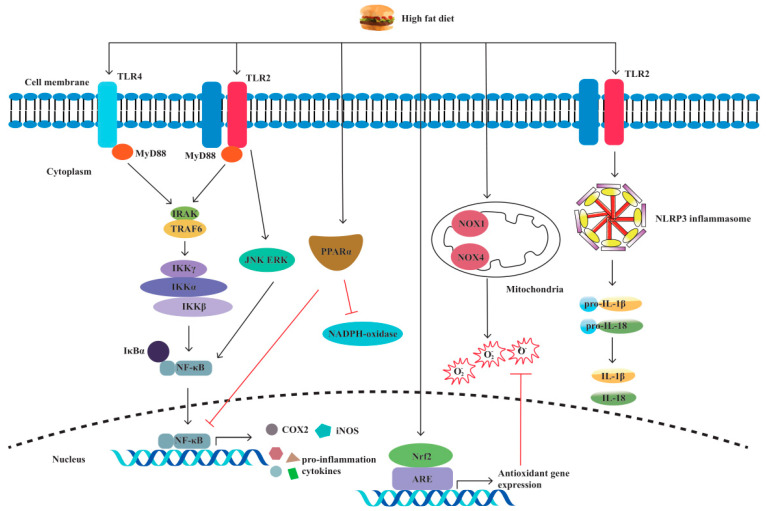
A high-fat diet regulates proinflammatory factors and oxidative stress through TLRs and NLRP. The specific manifestations are as follows: (1) Through the TLR4/NF−κB/MyD88 pathway, a high-fat diet (HFD) causes intestinal inflammation by elevating cytokine levels and alleviating cellular oxidative stress. (2) HFD activates oxidative stress markers nicotinamide adenine dinucleotide phosphate (NADPH), NADPH oxidase 4 (NOX4), and NOX1. (3) HFD increases the production of inducible nitric oxide synthase (iNOS) and cyclooxygenase-2 (COX-2). (4) HFD activates TLR2, resulting in M1 macrophage polarization and proinflammatory cytokine production. (5) HFD induces NLRP3 expression and promotes pro-IL-1β and IL-1β secretion by activating TLR2.

**Table 1 biomolecules-13-00905-t001:** Effects of the HF diet on intestinal immunity.

Effects of HFD on Intestinal Products or Microbiota	Mechanisms of Intestinal Products or Microbiota on the Intestine	References	Data Sources
AIEC	Targeting claudin-2 exacerbates the intestinal physical barrier and decreases the expression of the Muc-2 gene to disrupt the mucus barrier.	[143]	Mice
	Increasing TNF-α to active TNF-α-NF-κB regulatory pathways.	[144]	Mice
	Inducing Th17 polarization in the intestine.	[145]	Mice
	Eliciting the secretions of IL-8 and CCL20 to impact the recruitments of macrophages and dendritic cells in the intestine.	[146]	In vitro
*Clostridioides difficile*	Inactivating GTPases to lead cytoskeleton disruption and barrier breakdown.	[147]	Mice
	Inducing epithelial cell death and disrupting TJ structure.	[148]	In vitro
	Promoting NF-κB, IL-1 and IL-8 secretion.	[149]	Mice
	Decreasing neutrophil recruitment and modulating DCs and Th cell responses.	[150]	Mice
n-6 families	Reducing occludin secretion.	[118]	In vitro
	Promoting the antigen-presenting ability of DCs.	[124,125]	In vitro
	Decreasing the percentage of Th17 cells.	[125,127]	In vitro, Mice
*A. muciniphila*	Increasing the gene expressions of TLR2, TLR4, occludin, and claudin3 in the intestine promotes thickness of the mucus layer.	[151,152,153,154]	In vitro, In vitro, Mice, In vitro
	Enhancing the production of antimicrobial peptide Reg3γ.	[153]	Mice
	Inhibiting the expression of immunoglobulins encoding gene and receptor, chemokines CXCL13 and CCL12, and complement factors C1ra and C5ar1 in the colon.	[154]	In vitro
	Decreasing the presence of B cells in the colon and promoting Treg cell proliferation and antigen-specific Th cell response.	[155]	Mice
	Upregulating IL-8, IL-6, IL-1β, IL-10, and TNF-α secretion.	[156]	Mice
n-3 families	Protecting tight junctions while reducing MUC2 secretion.	[51]	Mice
	Upregulating the TLR-2 gene.	[119]	Mice
	Inhibiting neutrophil infiltration and averting the concomitant hurt caused by neutrophil production.	[120]	In vitro
	Reducing the antigen-presenting ability of DCs.	[121,122,157]	Mice, Mice, Rats
SCFAs	Stimulating MUC2 production.	[130]	In vitro
	Stimulating AMPs production.	[131]	Mice
	Promoting Th1 cells to produce IL-10 and DCs differentiation though mTOR and SATA3 pathways.	[132,134]	In vitro
	Increasing the antimicrobial functions of intestinal macrophages and downregulating macrophages secreting proinflammatory mediators.	[135,136]	Mice
	Inhibiting pro-inflammatory cytokines produced by neutrophils.	[138]	Rats

## Data Availability

Not applicable.

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
