# Peer review of "The Effects of a High-Fat Diet on Inflammatory Bowel Disease"

_biomolecules, 2023, doi:10.3390/biom13060905_

Round 1

Reviewer 1 Report

In the Manuscript “The effects of a high-fat diet on inflammatory bowel disease” by Dang et al., the authors describe in great details effects of high-fat diet (HFD) on inflammatory bowel disease (IBD). The authors performed extensive literature search and presented satisfactory literature review. The focus of the Manuscript is very interesting and clinically significant, yet it is not suitable for publication in its present form. Specific comments are following:

1.     The description or definition of IBD is missing. What are the symptoms? Why is it important to study it?

2.     The difference between Western diet and HFD (high fat diet) should be explained. There is an impression throughout the Manuscript that the authors are using the terms HFD and Western diet interchangeably.

3.     Lines 94-95 : should it be increasing intestinal permeability not “lessening” it?

4.     Lines 250-252: If NLRP3 inflammasome pro-inflammatory role has been discussed in great details, the same should be true for its protective role.

Reviewer 2 Report

Summary

In the article ‘The effects of a high-fat diet on inflammatory bowel disease’, Yuan Dang et al. describe current knowledge regarding mechanisms of dysregulation, inflammation and dysbiosis in the gut relating to high-fat and westernised diets. The review is very detailed and of interest to the scientific community but requires some changes before publication.

General comments

The majority of the review is based on mouse, rat and cell-line data. The authors should make sure that it is clear in the text what model system is referenced in each case, including in Table 1.

Although human data is more limited, you should address this and include a section/sections on the available human/clinical data.

In general, the figures need much more description in the legends and should be referenced more in the text where relevant.

Immune homeostasis in the gut should be described for the immune section– the balance of inflammatory and anti-inflammatory signalling.

The immune cell section is also lacking data about B cells (e.g. doi: 10.1189/jlb.0812394), ILCs (e.g. doi: 10.1172/jci.insight.99223) and NK cells (e.g. doi: 10.1039/D0FO02328K). 

Immunometabolism should also be addressed in this section, particularly in relation to fatty acids. Additionally, the effects of bacterial metabolites, such as SCFAs, on immune cells should be discussed.

Specific comments

Line 31: ‘excavate the reciprocal mechanisms’ – are the mechanisms reciprocal?

Line 36/36: ‘once intestinal immunity become dysfunctional’ – you are implying that immune dysfunction is the disease initiator, which isn’t known.

Line 45: ‘whereas such predominate decline’ – this does not make sense. Are you trying to say that these species are decreased in IBD?

Line 47/48: The citation here does not fully support the statement – what are their anti-inflammatory properties?

Line 49: should read ‘Adherent-invasive E. Coli (AIEC)’.

Line 54: What would be a personalised approach? Are you trying to say microbiota altering treatments would have to be personalised to patients?

Figure 1: The immune system also affects the intestinal barrier and IECs also influence microbiota.

Figure 2:  Needs a much more descriptive figure legend. Could include numbering in the figure to guide the reader.

Line 81: ‘posterior’ – it is not the posterior

Line 91: ‘zona occludin1 (ZO1)’ – zonula occludens-1 (ZO-1)

Line 92: add reference for TEER

Line 100: ‘drift of positive correlation’ – very vague. Was there a significant correlation or not?

Line 100/101 ‘intestinal paracellular’ – does not make sense

Line 105: add reference

Line 129: ‘evolve’ – differentiate

Line 133: ‘weaken’ – reduce

Line 134: ‘serum level’ – serum

Line 138-141: STAT3 is expressed by epithelial cells. What absence and impairment are you referencing?

Line 150: ‘… barrier integrity’ – reference

Line 151: ‘… than UC’ – reference

Line 161: ‘… occurrence’ - reference

Line 181: ‘… shown in previous research’ – reference

Figure 3: the figure legend needs a lot more detail to describe the figure.

Line 231: ‘… immune cells’ – reference

Line 246: ‘… NLRP3 deletion failed to recruit macrophages’ – there is no NLRP3 deletion in the referenced article.

Line 254: ‘… on IBD significance…’ – this doesn’t make sense.

Line 259: ‘NK T’ – NKT

Line 260: ‘higher rate’ – do you mean a higher ratio of Teff to Treg?

Line 264: ‘CD3+ T cells’ – CD3+ T cell infiltration in the gut

Line 267: ‘… in the colon.’ – compared to?

Line 268: More specifically what is meant by the ‘reprogramming of metabolites and function’?

Line 300: ‘Beguin et al.’ – reference

Table 1: ‘Inhabiting the expressions’ – Inhibiting the expression

Line 336: ‘… whose production’ – production of what?

Line 360: ‘… has been drowned’ – doesn’t make sense

Line 365: ‘… in charge of’ – capable of

Line 415: ‘… increases intestinal TEER’ – decreases

Line 419: ‘… promotes intestinal immune cell infiltration to decrease intestinal immunity’ – how is an increase in immune cells decreasing immunity? Do you mean they are promoting intestinal permeability and/or inflammation?

Line 422: ‘impertinent’ – I wouldn’t describe a diet as impertinent
